# Effect of the Morphology and Electrical Property of Metal-Deposited ZnO Nanostructures on CO Gas Sensitivity

**DOI:** 10.3390/nano10112124

**Published:** 2020-10-27

**Authors:** Sung-Ho Hwang, Young Kwang Kim, Seong Hui Hong, Sang Kyoo Lim

**Affiliations:** Division of Energy Technology of the Materials Research Institute, DGIST, 333 Techno Jungang-Daero, Hyeonpung-eup, Daegu 42988, Korea; hsungho@dgist.ac.kr (S.-H.H.); kimyk1211@dgist.ac.kr (Y.K.K.); kjyhsh@dgist.ac.kr (S.H.H.)

**Keywords:** CO sensor, ZnO, morphology, metal nanoparticle

## Abstract

The development of a highly sensitive gas sensor for toxic gases is an important issue in that it can reduce the damage caused by unexpected gas leaks. In this regard, in order to make the sensor accurate and highly responsive, we have investigated which morphology is effective to improve the sensitivity and how the deposited nanoparticle affects the sensitivity by controlling the morphology of semiconductor oxides—either nanorod or nanoplate—and depositing metal nanoparticles on the semiconductor surface. In this study, we compared the CO gas sensitivity for sensors with different morphology (rod and plate) of ZnO nanostructure with metal nanoparticles (gold and copper) photodeposited and investigated the correlation between the gas sensitivity and some factors such as the morphology of ZnO and the properties of the deposited metal. Among the samples, Au/ZnO nanorod showed the best response (~86%) to the exposure of 100 ppm CO gas at 200 °C. The result showed that the electrical properties due to the deposition of metal species also have a strong influence on the sensor properties such as sensor response, working temperature, the response and recovery time, etc., together with the morphology of ZnO.

## 1. Introduction

Carbon monoxide (CO) is a dangerous byproduct derived from the incomplete combustion of petroleum-based fuels. It is very harmful to the human body because it prohibits carrying oxygen to the heart and brain resulting from the rapid formation of carboxyhemoglobin in the lungs [1]. According to the WHO reports, it has been reported that humans die within an hour when exposed to 800 ppm of carbon dioxide, and the recommended safety level is 30 ppm to exposure of CO for an hour [2]. These standards call for an accurate and rapidly responsive sensor for CO gas. However, current CO sensors require a high-temperature condition that reduces the durability to ensure high sensitivity [3]. It is a fascinating issue for scientists to develop a highly sensitive CO gas sensor that not only detects the extremely low concentration of CO rapidly, but also operates at a low temperature. Therefore, a considerable number of papers have been reported on CO gas sensors using semiconducting materials such as ZnO [4], In_2_O_3_ [5,6], SnO_2_ [3], etc. Among them, ZnO is one of the best materials for the CO gas sensor due to its non-toxic nature, and chemical and thermal stability. Therefore, a considerable study has been reported about the synthesis and application of the CO gas sensor in the form of ZnO nanostructures (i.e., rod, sphere, and plate), p-type semiconductor/ZnO, metal-deposited ZnO composites, etc. However, there have been few comparative studies relating to the effect of the morphology of ZnO on CO gas sensitivity [7,8]. According to the previous reports, the sensitivity toward CO gas is highly governed by the intrinsic defects of ZnO such as oxygen vacancies and zinc interstitials. It is well known that these defects act as more favorable active sites to adsorb oxygen species or CO molecules on the surface [9]. The dopant also affects the response of the gas by variation of the electrical properties of the ZnO. However, it is not fully understood which metal element has a strong influence on CO gas sensitivity. Although there are several works reported about several synthetic routes to prepare the metal/semiconductor gas sensors via chemical doping [10] or chemical reduction of metal ion [11,12,13] as listed in Table 1, to our knowledge, there are few studies reporting on metal/semiconductor gas sensors prepared via photodeposition. The photodeposition method has some advantages compared with other methods, in that metal nanoparticles could be not only deposited on the substrate in uniform size and distribution by photodeposition, but also the fabrication process would be easier and more simple than other synthetic routes. The electrical property of the metal nanoparticles might be one of important variables to affect the CO gas response. In order to investigate such effect, generally noble metal would be considered as the best candidates due to their high electrical conductivity and stability. Furthermore, in this work, considering practical application, we wanted to find other metal of high electrical conductivity and stability with the price competitiveness to alternate noble metals. In this aspect, altogether with Au nanoparticle, Cu one was chosen as one of the candidate materials. Interestingly, it was reported that both Au and Cu nanoparticles have important role in CO sensing [14,15]. The presence of Au nanoparticles on the surface of In_2_O_3_ nanowire serves to enhance the CO oxidation due to a higher oxygen ion chemisorption on the conductive Au nanoparticles surfaces. The coexistence of ZnO and Cu metal could enhance the capability of sensor material to adsorb CO molecules as the copper has active sites to adsorb CO molecules at both low and high temperature [15]. Therefore, in this study we prepared metal-deposited ZnO nanorods and nanoplates via photodeposition. Then, we compared the CO gas sensitivity of each metal-deposited ZnO nanostructure. The correlation between CO gas sensitivity and some possible factors, such as morphology of semiconductor, electrical properties, intrinsic properties of metal, and surface characteristics, was discussed.

## 2. Materials and Methods

### 2.1. Synthesis of ZnO Nanostructures

ZnO nanorods were synthesized by the solvothermal method [16]. First, a zinc acetate dihydrate solution of 0.5 M (Zn(CH_3_COOH)_2_•2H_2_O, 99%, Daejung, Korea) was prepared by stirring in ethanol for 10 min. Then, sodium hydroxide (NaOH, 99%, Daejung, Korea) was added to the solution so that the concentration of NaOH was 5 M, followed by stirring for 30 min. After transferring the mixed solution to an autoclave, the solution was stirred at 150 °C for 2 h and cooled down to room temperature. The precipitates were then washed with distilled water until the pH of precipitates became neutral and dried in a vacuum oven at 70 °C for 12 h. ZnO nanoplates were synthesized by the same method of the ZnO nanorods except for changing the zinc precursor from Zn(CH_3_COOH)_2_•2H_2_O to zinc chloride (ZnCl_2_, 99%, Daejung, Korea). Then, uniform ZnO nanorods and nanoplates were obtained (Appendix A).

### 2.2. Photodeposition of Metals on ZnO Nanostructures

The metal-deposited ZnO nanostructures were prepared by the photodeposition method [17,18]. Briefly, ZnO nanorods or ZnO nanoplates (0.5 g L^−1^) were dispersed in 20 wt% methanol solution Then, the metal precursor was added into the mixture solution so that the concentration of metal was 0.1 mM. The metal precursor for gold and copper was selected as gold chloride (AuCl_3_, Sigma-Aldrich, St. Louis, MO, USA) and copper acetate monohydrate (Cu(CH_3_COOH)_2_•H_2_O, Sigma-Aldrich, St. Louis, MO, USA), respectively. It was found that the crystal phase of ZnO nanostructure was not affected by UV irradiation to the suspension as shown in XRD patterns of Appendix A. However, the amount of metals deposited on the ZnO nanostructures would be increased with increasing the UV intensity and deposition time, and then saturated. As the gas sensing response of metal/semiconductor sensor would be closely related to the amount of deposited metal, the effect of UV intensity and deposition time (irradiation time) upon the amount of metal deposited on ZnO nanostructures was checked using 0.1 mM of metal precursors to optimize the deposition amount of each metal. The deposition amount of metals increased with increasing the UV intensity up to 450 W for 30 min in deposition process. The content of deposited metal was observed to be saturated at condition over 30 min. On the basis of this result, we set the experimental condition for the deposition of metal as UV intensity of 450 W for 30 min in order to obtain the best gas response throughout the optimal loading content of metal nanoparticles deposited on ZnO nanostructures. Therefore, after transferring the mixed solution into a cylindrical quartz reactor, a UV light (λ = 254 nm, 450 W) was irradiated to a mixed solution for 30 min. The resultant precipitates were washed with distilled water and dried at room temperature for 24 h. Then, metal deposited ZnO nanostructures were obtained and characterized.

### 2.3. Surface Characterization

High-resolution morphological images of the ZnO nanostructures and metal deposited ZnO nanostructures were obtained using a scanning electron microscope (SEM, SU8020, Hitachi, Tokyo, Japan) and transmission electron microscope (TEM, HF-3300, Hitachi, Tokyo, Japan). The crystal structures of the samples were analyzed using an X-ray diffractometer (Empyrean Alpha-1, Malvern Panalytical Ltd., Malvern, UK) with Cu Kα radiation (λ = 1.54178 Å) and operating at 40 kV, 30 mA, and a scan rate of 0.03 s^−1^. The chemical states of the samples were analyzed by an X-ray photoelectron spectrometer (Escalab 250 Xi, Thermo Fisher Scientific, Waltham, MA, USA). The deposition amount of inorganic elements in the samples was analyzed by inductively coupled plasma mass spectrometer (iCAP7400DUO, Thermo Fisher Scientific, Waltham, MA, USA). A photoluminescence experiment was performed with a 325 nm He-Cd laser (1 K, 50 mW, Kimmon Koha Co. Ltd., Tokyo, Japan). The Brunauer–Emmett–Teller (BET) surface area of ZnO nanostructures was calculated from nitrogen adsorption–desorption isotherm at 77 K (ASAP2020, Micrometrics, Norcross, GA, USA). The effective surface areas were estimated at a relative pressure (P/P_0_) ranging from 0.06 to 1. The electrical conductivity of the samples was measured using a 4-point probe system (CMT-SR200N, AIT, Suwon, Korea).

### 2.4. Measurement of CO Gas Sensitivity

The sensor was fabricated by dropping ZnO nanostructures or metal deposited ZnO nanostructures gels on a sapphire substrate with pre-patterned Pt electrodes (Figure 1). The gels were made by grinding 10 mg of ZnO nanostructures or metal-deposited ZnO nanostructures with 0.25 mL of distilled water in an agate mortar. A precise micropipette (Eppendorf Reference^®^ 2, Eppendorf, Germany) was then used to deposit the solution. After dropping 10 μL of ZnO nanostructures or metal-deposited ZnO nanostructures, the substrate was dried in the air [19]. The CO gas sensitivity of the as-prepared sensors was measured by a computer-based sensing system consisting of three parts: (i) a controlling device of gas (PXI-DAQ system, National Instrument, Austin, TX, USA) equipped with a mass flow controller (Brooks 5850E, Brooks instrument, Hatfield, PA, USA), (ii) a semiconductor characterization system (4200SCS, Keithley, Cleveland, OH, USA), and (iii) a chamber (size: 20 cm × 20 cm × 10 cm) equipped with a heater for sensors (E3631A, Agilent Technologies, Inc., Wood dale, IL, USA). The gas response was defined as the percentage change in resistance of the sensor upon exposure of CO gas ((R_a_ − R_g_)/R_a_) × 100 (%), where R_a_ and R_g_ are the resistance in air and CO, respectively.

## 3. Results and Discussion

### 3.1. Surface Characterization of ZnO Nanostructures and Metal-Deposited ZnO Nanostructures

The local microstructure of all the samples were showed in Figure 2, in which nanorods were much smaller than nanoplates and very small metal nanoparticles such as copper and gold nanoparticles were observed to be deposited on ZnO nanorods and nanoplates. In the case of the overall morphology, both ZnO nanorods and nanoplates were observed to exhibit uniform overall morphology and distribution (SEM images of Appendix A). On the other hand, to investigate the phase of metal-deposited ZnO nanostructures, we carried out XRD and SAED patterns analysis. While the XRD in Appendix A showed that only the crystal phase of ZnO alone was observed in all the samples due to the low crystallinity of small metal nanoparticles (~10nm) and smaller content of deposited metal for that of ZnO, it could be confirmed that, from the TEM images and onset SAED patterns of Figure 2, the deposited metal species were stabilized on ZnO nanostructures as metal phase. On the other hand, it seemed in Figure 2 that a larger amount of nanoparticles was deposited on the ZnO nanoplate than on the ZnO nanorod. It was also shown that the amount of metal deposited on ZnO nanostructures were varied according to metal species. The exact amount of deposited metal on ZnO nanostructures was confirmed by ICP-MS measurement (Table 2). The molar ratios of Au/Zn in Au/ZnO nanostructures were calculated to be 0.327 and 0.497 for nanorods and nanoplates, respectively, while those of Cu/Zn in Cu/ZnO nanostructures were 0.061 and 0.084 for nanorods and nanoplates, respectively. This showed that all the metals were more largely deposited on the ZnO nanoplates than nanorods as well as Au nanoparticles were much more largely deposited than Cu nanoparticles in both ZnO nanorod and nanoplate. These results could be understood due to different electrostatic interactions or weak van der Waals forces at the surface of the samples induced by the morphology of ZnO and the electrical property of metal species [20]. In other words, the amount of deposited-metal was closely related to the pH of the metal precursor suspension and the surface charge of the ZnO nanostructure at that pH [21,22]. In Figure 3a, we present the Zeta potential of ZnO nanostructures in metal precursor solution (black) and pH of metal precursor solution (gray). The ZnO suspensions with Au precursor were the most basic, and the surface charge of ZnO has the greatest negative charge among the ZnO suspensions with metal precursors, which enabled strong interaction with metal cations. Therefore, it could be expected that the gold nanoparticles would be more largely deposited than Cu nanoparticles. It was also notable in Table 1 that the electrical conductivity was higher in metal-deposited ZnO nanorods than in metal-deposited ZnO nanoplates due to the combined effect of the metal deposited amount and morphology of ZnO. The XPS Au 4f spectra of Au/ZnO nanostructures and the XPS Cu 2p spectra of Cu/ZnO nanostructures are shown in Figure 3b,c. Two metallic Au 4f peaks with the binding energies at 84.0 and 87.7 eV corresponding to Au 4f_7/2_ and Au 4f_5/2_, respectively, were observed in the XPS spectra of metal-deposited samples with gold chloride [23]. In the case of copper acetate monohydrate, two metallic Cu 2p peaks were observed in the samples with binding energies at 932.55 and 952.55 eV corresponding to Cu 2p_3/2_ and Cu 2p_1/2_, respectively [18,24]. Therefore, from the ICP and XPS results, it was confirmed that the metallic nanoparticles (Cu^0^ or Au^0^) were well deposited on each ZnO nanostructure via the photodeposition method.

### 3.2. CO Gas Sensitivity of ZnO Nanostructures and Metal-Deposited ZnO Nanostructures

Figure 3 shows the gas responses of the pristine ZnO nanostructures and metal-deposited ZnO nanostructures to a 100 ppm of carbon monoxide as a function of working temperature. It is observed that the responses of the pristine ZnO nanorod and nanoplate increase with an increase of the working temperature up to 300 °C, then decrease with a further increase in working temperature. The primary oxygen species on the surface of pristine ZnO nanostructure can be known to be changed by temperature. Monovalent oxygen anions (O_2_^−^ and O^−^) can exist at below 300 °C. However, divalent oxygen anion (O^2−^), which induces larger resistance changes via a two-electron reaction, can exist at over 300 °C [25]. Simultaneously, the desorption of oxygen species increases with a further rise in temperature. Therefore, the optimal working temperature of the pristine ZnO nanostructures can be decided by the combined effect of two factors, the formation of divalent oxygen anion and the rate of adsorptive reaction, at a certain temperature. However, the optimal working temperature becomes lower to 200 °C in all the metal-deposited ZnO nanostructures. It is attributed to the lowering of the activation energy for chemisorption of CO gas by metals [11].

On the other hand, in Figure 4, Figure 5 and Table 3, the Au/ZnO nanorods exhibit the best response among all the samples. The Cu/ZnO nanorods show an intermediate response. The lowest response is observed in the pristine ZnO nanoplates. Compared with the response of the ZnO nanorods, the responses of the ZnO nanoplates are observed to be slightly decreased, and the response is increased in the following order; Au/ZnO nanoplate, Cu/ZnO nanoplate, and ZnO nanoplate, which is the same increasing order of responses in ZnO nanorods. That is, it has been observed that the CO gas response is higher in ZnO nanorods than in ZnO nanoplates due to the difference in morphology. As shown in Figure 2, ZnO nanorods would be expected to have a higher surface area than nanoplates with a much larger particular size. In Figure 6 and Table 4, it could be confirmed that nanorod morphology showed a higher relative surface area and larger pore volume due to their mesoporosity than nanoplate one. In addition, it could become a good reason for a higher response of nanorod morphology that the high aspect ratio and interconnectivity of the nanorods would result in better electron transport along the axial direction and the creation of more electron paths [26]. Actually, it was found that the electrical conductivity of nanorods was higher than that of corresponding nanoplates in Table 1.

On the other hand, some factors were also selected to identify the reasons for the improvement of gas sensitivity in metal-deposited ZnO nanostructures relative to the pristine ZnO nanostructures. The loading amount of metal, a metallic component, and the work function of metal might influence the sensitivity of the sensors. Therefore, we investigated the correlation between those factors and sensitivity. The electrical properties of the sample can affect the adsorption of the target gas ions on the surface of the sample, resulting in the change of the resistance related to the gas sensitivity of the sample. Therefore, in this work, we will discuss the relationship between the electrical conductivity, the work function of the samples, and their gas sensitivity [27]. ZnO is an n-type semiconductor that contains free electrons, and, after photodeposition of metal on the ZnO nanostructures, free electrons from the metal are released into the ZnO structure and the number of free electrons will be greater [28]. According to the principle of CO gas sensing, the oxygen species are adsorbed on the surface of the n-type semiconductor gas sensor, which results in a high resistance in the atmosphere due to the formation of a depletion layer. After exposure to CO gas, the adsorbed oxygen species reacted with CO molecules, which releases the free electrons back to the ZnO structure, and brings about a resistance change of the sample, and then the level of resistance change determines the response of the sensor. It is reported that the high electrical conductivity and work function of metal can promote the adsorption rate of the exposed gas molecules and the level of resistance changes. The response of the sensor is related to the creation of the Schottky barrier between the metal clusters and ZnO grains. In the ambient air, oxygen molecules are adsorbed on the metal clusters, increasing their work function and decreasing their electrical conductivity. On the contrary, in the presence of CO gas, desorption of oxygen molecules from the deposited metal surface leads to a decrease in the work function and an increase in electrical conductivity. According to the gas-sensing principle, we can predict that a better response can be achieved by inducing a larger change in the work function or electrical conductivity of the sensor under the exposure of air and CO. From this point of view, the metal-deposited ZnO nanostructures, due to the larger electrical conductivity and higher work function of deposited-metal relative to ZnO, could bring about a larger difference in the height of the Schottky barrier, leading to an improved gas response [29,30]. It was found in Table 1 that the electrical conductivity of the samples became higher in the order of Au-deposited ZnO nanostructures, Cu-deposited ones, and pristine ones due to the actual fraction of metal-deposited on ZnO nanostructures. The work function of the deposited metal also increased in the order of Au (5.10–5.47 eV) and Cu (4.53–5.10 eV), which was reported elsewhere. As shown in Figure 7, due to the higher work function of Au relative to ZnO, electrons migrate from the conduction band of ZnO to Au in order to equalize the Fermi levels and form Schottky heterojunctions at the interface of Au and ZnO nanostructures, which led to the reduction of conduction volume. Such reduction of conduction volume will contribute to the enhanced sensor response in Au/ZnO [29]. Analogous to the case of Au/ZnO, such reduction in Cu/ZnO will also occur but be smaller than that in Au/ZnO due to the lower work function of Cu relative to Au. Therefore, we can expect that the Au-deposited ZnO nanorods, which have the highest electrical conductivity and work function of deposited-metal, would elicit the best response to the exposure of CO gas.

In general, it is well known that the surface oxygen of the sample also influences the gas response. Therefore, we analyzed the amount of surface oxygen of the sample by XPS analysis. To estimate the fraction of each component of surface oxygen, the XPS O1s spectra were deconvoluted using the Shirley method. Figure 8 shows the XPS O1s spectra of the ZnO nanostructures and metal-deposited ZnO structures. It was found that the samples were composed of three kinds of surface oxygen with different fractions. The peaks in XPS O1s spectra assigned at 530.3, 531.2, and 532.6 eV were lattice oxygen, oxygen ions in the oxygen-deficient regions caused by oxygen vacancy, and chemisorbed oxygen, respectively [31,32]. As shown in Table 5, it was found that the peak at ~531.2 eV attributed to oxygens in oxygen-deficient regions oxygen vacancies was larger in the ZnO nanorods than in the ZnO nanoplates, reflecting more oxygen vacancies in ZnO nanorod than in ZnO nanoplate. In addition, because of the larger peak area in metal-deposited ZnO nanostructures than pure ones, it was considered that the deposition of metal nanoparticles also increases the creation of the oxygen vacancies in the samples. Therefore, it was confirmed that the morphology of ZnO nanostructures and the deposition of metal on the ZnO nanostructures affect the amount of surface oxygen of the samples. Among the samples, Au/ZnO nanorod exhibited the largest portion of oxygen vacancies, which was also believed to have contributed to the response to the CO gas.

The response time and stability of the sensor are also important factors to the practical application of the sensor [33,34]. Therefore, we measured the response time of the sensors by repeating exposure to CO and air in one cycle. Figure 9 shows the representative actual response transient, response time, and recovery time of the as-prepared sensors. The response time and recovery time are listed in Table 3. In pristine ZnO nanostructures, the response time is up to ~200 s. However, that of metal-deposited ZnO nanostructures is below 35 s. This result was also observed similarly in recovery time and it was found that the response and recovery times were observed to be much shorter in the metal-deposited ZnO nanostructures than in pristine ones. Therefore, the deposition of metals such as Au and Cu might have an influence on the significant reduction of both the response and recovery time.

The reproducibility test was also carried out using metal-deposited ZnO nanostructures and pristine ZnO nanostructures under the reversible exposure to 100 ppm CO gas at 200 °C as shown in Figure 9. It was found that the response, response time, and recovery time of the metal-deposited ZnO nanostructures were almost identical during 10 reversible cycles. As shown in Figure 10, we also investigated the change of chemical state of metal in the metal-deposited ZnO nanostructures after 10 successive cycles of exposure to 100 ppm CO gas. Au 4f and Cu 2p spectra of samples before and after gas-sensing test did not show any prominent difference, reflecting that there was no change in the chemical state of Au and Cu nanoparticles after gas-sensing test. Those results would also indicate that the metal-deposited ZnO nanostructures have good stability.

## 4. Conclusions

In this study, we prepared metal-deposited ZnO nanorod and nanoplate via photodeposition. The CO gas sensitivity is influenced by the combined effect of electrical property such as the work function of deposited-metal and electrical conductivity, the amounts and type of the deposited-metal nanoparticles, and the morphology of the ZnO nanostructures in metal-deposited ZnO nanostructures. Among the samples, Au/ZnO nanorod showed the best response (~86%) to the exposure of 100 ppm CO gas at 200 °C. The result showed that the electrical properties due to the deposition of metal species also have a high influence on the sensor properties such as sensor response, working temperature, the response and recovery time, etc., together with the morphology of ZnO. These findings could be expected to enlarge the understanding of the role of the morphology of sensors and deposited-metal on gas sensitivity.

## Figures and Tables

**Figure 1 nanomaterials-10-02124-f001:**
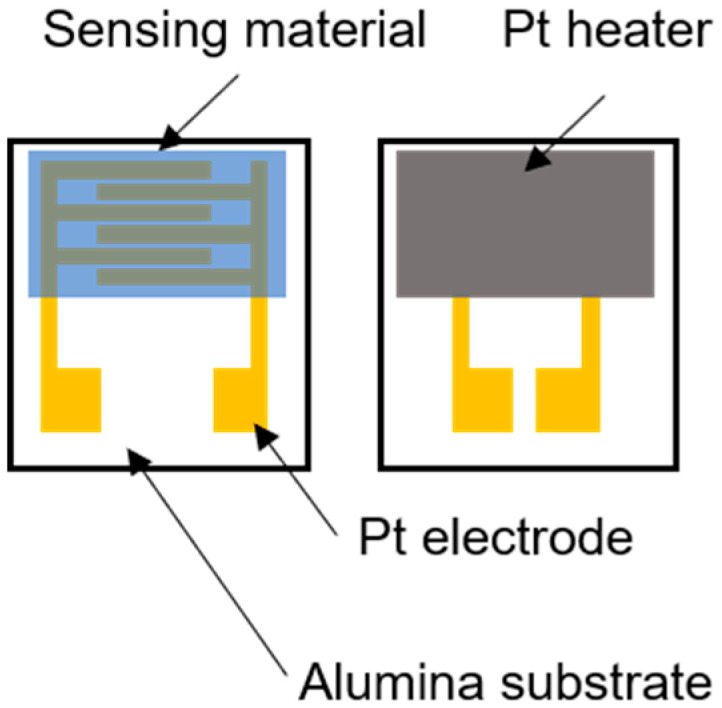
The schematic illustration of the sensor device.

**Figure 2 nanomaterials-10-02124-f002:**
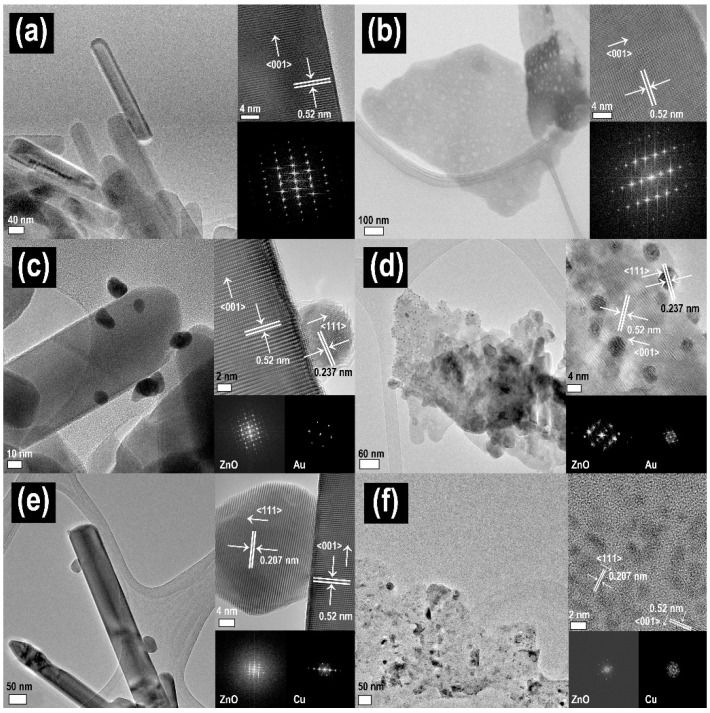
TEM images and selected area electron diffraction patterns of metal-deposited ZnO nanostructures: (**a**) ZnO nanorod, (**b**) ZnO nanoplate, (**c**) Au/ZnO nanorod (**d**) Au/ZnO nanoplate, (**e**) Cu/ZnO nanorod, and (**f**) Cu/ZnO nanoplate.

**Figure 3 nanomaterials-10-02124-f003:**
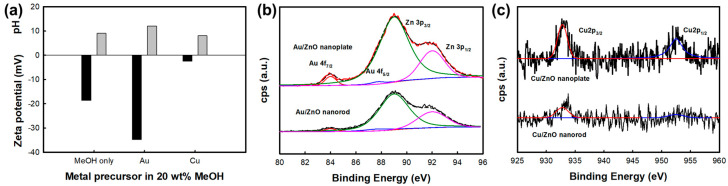
(**a**) Zeta potential of ZnO nanostructures in metal precursor solution (black bar) and pH of metal precursor solution (gray bar). The concentration of the metal precursor is 0.1 mM, (**b**) XPS Au 4f spectra of Au/ZnO nanostructures, and (**c**) XPS Cu 2p spectra of Cu/ZnO nanostructures.

**Figure 4 nanomaterials-10-02124-f004:**
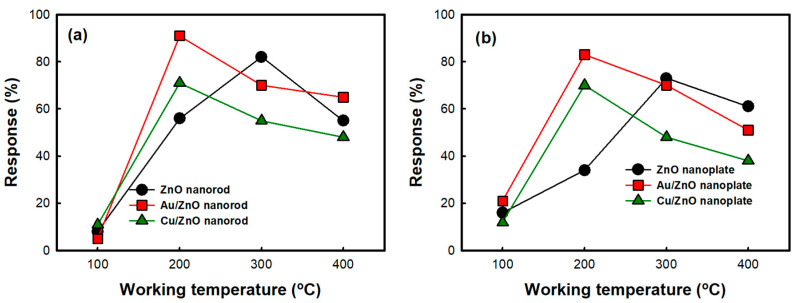
The gas response of the samples up to 100 ppm CO as a function of working temperature (**a**) ZnO nanorod and metal-deposited ZnO nanorods (**b**) ZnO nanoplate and metal-deposited ZnO nanoplates.

**Figure 5 nanomaterials-10-02124-f005:**
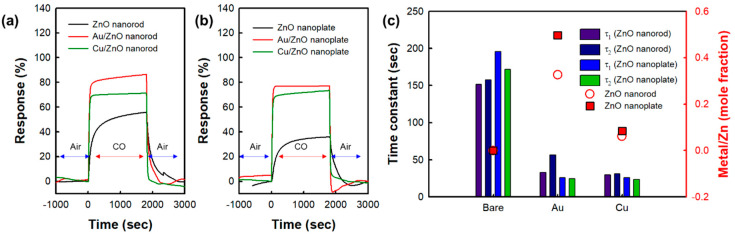
The actual response transient of pristine ZnO nanostructures and metal-deposited ZnO nanostructures to 100 ppm CO gas at 200 °C. (**a**) ZnO nanorod and metal-deposited ZnO nanorods. (**b**) ZnO nanoplate and metal-deposited ZnO nanoplates. (**c**) Response and recovery time of pristine ZnO nanostructures and metal-deposited ZnO nanostructures.

**Figure 6 nanomaterials-10-02124-f006:**
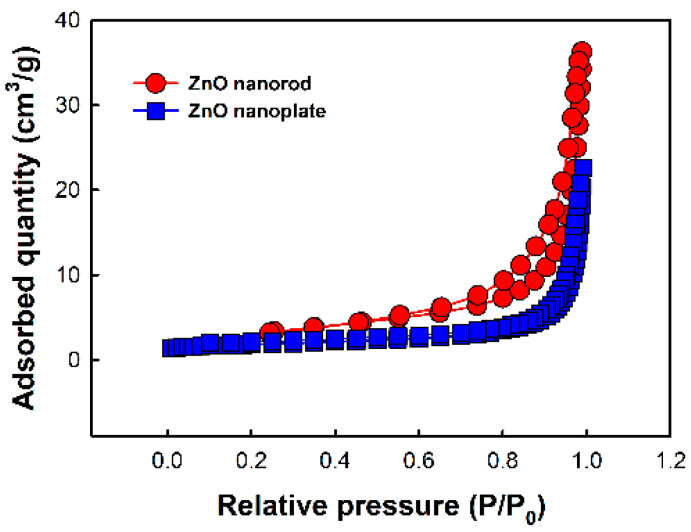
The Brunauer–Emmett–Teller (BET) surface area measured from N_2_ adsorption and desorption isotherms at 77 K.

**Figure 7 nanomaterials-10-02124-f007:**
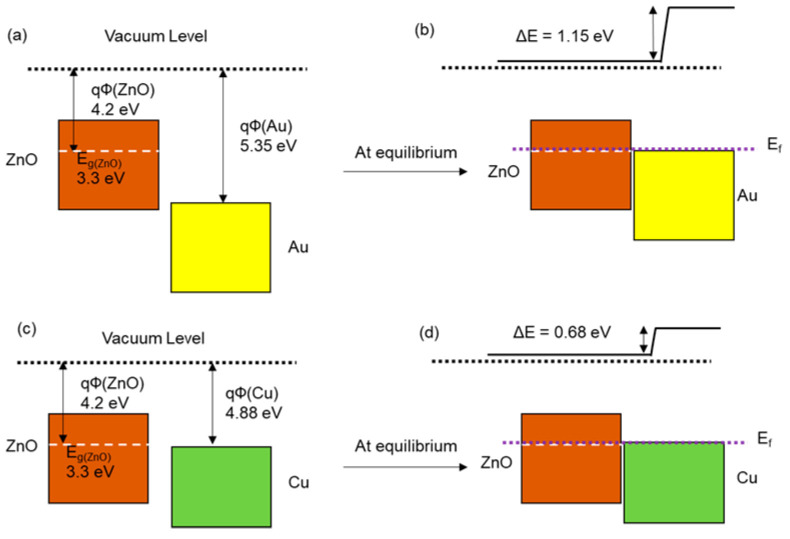
(**a**) Energy levels of ZnO and Au; (**b**) formation of heterojunction barrier. (**c**) Energy levels of ZnO and Cu; (**d**) formation of heterojunction barrier.

**Figure 8 nanomaterials-10-02124-f008:**
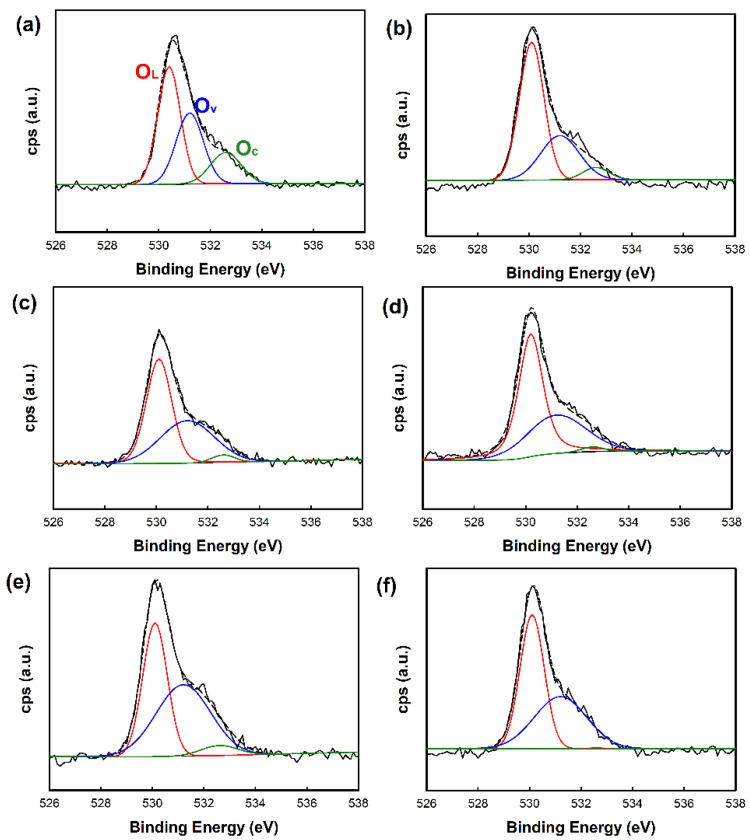
XPS O1s spectra of the ZnO nanostructures and metal-deposited ZnO nanostructures: (**a**) ZnO nanorod, (**b**) ZnO nanoplate, (**c**) Au/ZnO nanorod, (**d**) Au/ZnO nanoplate, (**e**) Cu/ZnO nanorod, and (**f**) Cu/ZnO nanoplate.

**Figure 9 nanomaterials-10-02124-f009:**
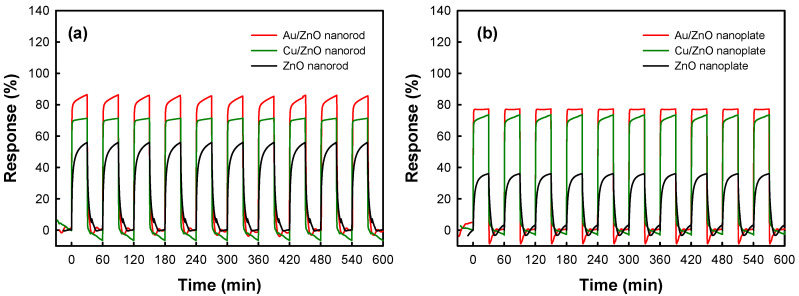
Gas response of the ZnO nanostructure and metal deposited ZnO nanostructures during 10 successive cycles of exposure to 100 ppm CO and air alternately at 200 °C. (**a**) ZnO nanorod and metal-deposited ZnO nanorods (**b**) ZnO nanoplate and metal-deposited ZnO nanoplates.

**Figure 10 nanomaterials-10-02124-f010:**
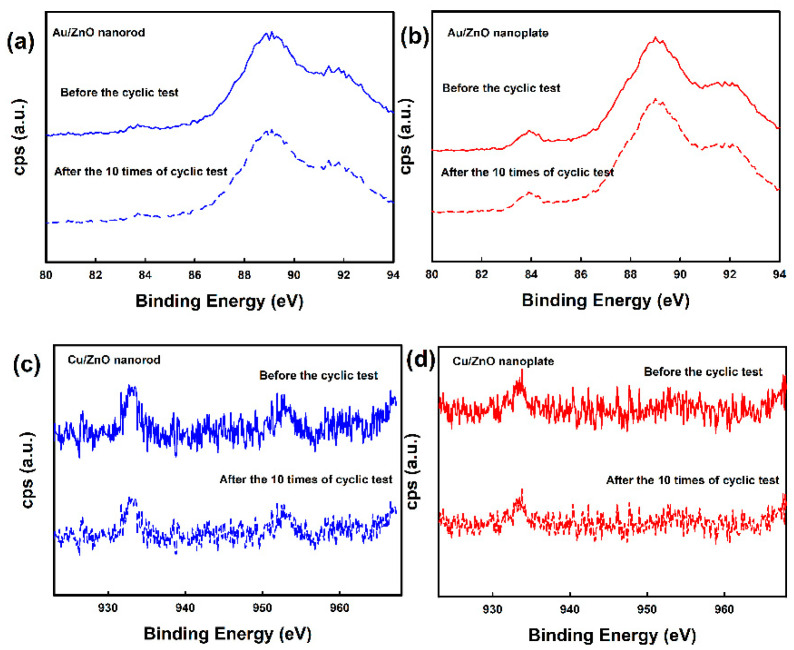
The chemical states of metal deposited ZnO structures before and after the stability test: (**a**) Au 4f spectra of Au/ZnO nanorod, (**b**) Au 4f spectra of Au/ZnO nanoplate, (**c**) Cu 2p spectra of Cu/ZnO nanorod, and (**d**) Cu 2p spectra of Cu/ZnO nanoplate.

**Table 1 nanomaterials-10-02124-t001:** Comparison of the CO sensitivity of metal/semiconductor gas sensors with different synthetic routes.

Nano Particles	Morphology (ZnO)	Synthesis Method	Operating Temperature (°C)	Concentration of CO (ppm)	Sensitivity (R_a_/R_g_)	Ref.
In	Nanoparticle	Sol–gel	300	100	5	[10]
Au	Nanostar	Hydrothermal	RT	100	15	[12]
Pt	Nanosheet	Hydrothermal/calcination	180	100	3.5	[11]
Au	Nanorods	Hydrothermal	150	1000	12	[13]
Pd	Nanowires	VLS growth	RT	0.1	1.02	[13]

**Table 2 nanomaterials-10-02124-t002:** The elemental ratio in different metal-deposited ZnO nanostructures and the electrical conductivity of the metal-deposited ZnO nanostructures.

Sample	Metal (mM)	Zn (mM)	Metal/Zn ^1^	Electrical Conductivity (S/m)
ZnO nanorod	-	3.313	-	7.345 × 10^−7^
Au/ZnO nanorod	1.040	3.186	0.327	6.327 × 10^−6^
Cu/ZnO nanorod	0.185	3.048	0.061	1.057 × 10^−6^
ZnO nanoplate	-	3.061	-	6.148 × 10^−7^
Au/ZnO nanoplate	1.501	3.019	0.497	5.113 × 10^−6^
Cu/ZnO nanoplate	0.254	3.030	0.084	1.044 × 10^−6^

^1^ The molar ratio of the photodeposited metal (Au or Cu) vs. zinc.

**Table 3 nanomaterials-10-02124-t003:** The gas response, response time, and recovery time of the samples.

Sample	^1^ Gas Response (%)	^2^ Response Time τ_1_ (sec)	^3^ Recovery Time τ_2_ (sec)
ZnO nanorod	55.96	151.519	157.970
Au/ZnO nanorod	86.45	32.864	56.599
Cu/ZnO nanorod	71.39	29.851	31.108
ZnO nanoplate	36.06	195.691	172.005
Au/ZnO nanoplate	77.34	26.053	24.749
Cu/ZnO nanoplate	70.51	26.172	23.743

^1^ Gas response of the sample at the working temperature of 200 °C, ^2^
$\Delta R=\Delta R_{max}\;(1-e^{-\frac{t}{\tau_{1}}}$), ^3^
$\;\Delta R=\Delta R_{max}\;e^{-\frac{t}{\tau_{2}}}.$

**Table 4 nanomaterials-10-02124-t004:** BET surface area and pore characteristics.

Sample	BET Surface Area (m^2^/g)	Pore Volume (cm^3^/g)	Pore Size (nm)
ZnO nanorod	11.3021	0.040563	12.945
ZnO nanoplate	5.9611	0.034951	23.453

**Table 5 nanomaterials-10-02124-t005:** Surface oxygen components in ZnO nanostructures and metal-deposited ZnO nanostructures.

Sample	O_L_^1^(%)	O_v_^2^(%)	O_c_^3^(%)
ZnO nanorod	39.63	46.39	13.98
Au/ZnO nanorod	43.49	51.66	4.85
Cu/ZnO nanorod	52.78	43.91	3.31
ZnO nanoplate	62.59	31.47	5.94
Au/ZnO nanoplate	50.00	48.31	1.69
Cu/ZnO nanoplate	54.20	45.60	0.20

^1^ lattice oxygen; ^2^ oxygen around vacancy; ^3^ chemisorbed oxygen.

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
