# Peer review of "Effect of the Morphology and Electrical Property of Metal-Deposited ZnO Nanostructures on CO Gas Sensitivity"

_nanomaterials, 2020, doi:10.3390/nano10112124_

Round 1
Reviewer 1 Report
The article undoubtedly contains very interesting results that are suitable for publication in the Sensors. However, before publication, some experimental data should be added and discussion of the results should be strengthened.
- The Materials and methods section indicates that x-ray diffraction studies have been performed. These data should be added to the discussion of whether UV irradiation of the suspension affects the size of zinc oxide crystallites.
- The paper uses the XPS method. The Au and Cu spectra are discussed and the conclusion about the electronic state of these elements is made. Obviously, the authors should have similar data for oxygen (O1s). It is necessary to add these results to the article and discuss how the amount of surface oxygen changes depending on the microstructure of zinc oxide particles and the presence of deposited copper and gold particles.
- In addition, it would be useful to make a blank experiment in which the conditions for the deposition of metal particles would be reproduced, without introducing metal precursors. The ZnO particles treated in this way should also be examined using the XPS method. In this way, it is possible to evaluate the effect of UV radiation directly on the characteristics of zinc oxide particles.
- The authors presented very interesting results on the dynamic characteristics of sensors - a significant reduction in response time and recovery time in the case of modified samples compared to pure ZnO. This fact requires a deeper discussion.
- The title of the article is "Effect of the morphology...", but in fact this effect is only postulated on the basis of experimental data, but there are no theoretical or model explanations for this effect in the work. This is the weak point of the reviewed article.
Author Response
Dear Reviewers
First of all, I’d like to express my sincere and thanks to Reviewers who read my manuscript with deep concerns and to valuable comments.
According to reviewer’s comments, manuscript was revised and checked to avoid some typical mistakes and expression or notation to cause misunderstanding.
We appreciate kind comments and suggestion of reviewer again. Our responses to these comments are listed below, and the manuscript have been accordingly revised. All the changes were represented in blue color in revised manuscript.
Thank you for your kindness and help
Sincerely yours
Response to Reviewer 1 comments
The article undoubtedly contains very interesting results that are suitable for publication in the Sensors. However, before publication, some experimental data should be added and discussion of the results should be strengthened.
Comment 1. The Materials and methods section indicates that x-ray diffraction studies have been performed. These data should be added to the discussion of whether UV irradiation of the suspension affects the size of zinc oxide crystallites.
Response 1.
According to reviewer’s comment, we added X-ray diffraction data in supplementary section and the related discussion in experimental section. As shown in the Materials and methods section, firstly, we synthesized the ZnO nanostructures (rod and plate) via hydrothermal method. Then, we prepared metal-deposited ZnO nanostructures via photodeposition with different metal precursors on the as-prepared ZnO nanostructures. According to the XRD analysis of Figure S2 and TEM images of Figure 2, the crystal phase and size of zinc oxide crystallites seemed to be rarely affected by UV irradiation of the suspension. Instead, in this work, we examined the effect of UV intensity and deposition time (irradiation time) on the content of deposited-metals on the ZnO nanostructures to optimize the deposition amount of each metal. The deposition amount of metals increased with increasing the UV intensity up to 450 W for 30 min. The content of deposited metal was observed to be saturated at the condition of deposition time over 30 min. The gas sensitivity is proportional to the amount of deposited metals. Therefore, We set the experimental condition for the deposition of metal as UV intensity of 450 W for 30 min in order to obtain the best gas response throughout the optimal content of metal nanoparticles deposited on ZnO nanostructures.
Related discussions are in the experimental section (2. Materials and Methods) as follows.
Page 3, Line 90
~ respectively. It was found that the crystal phase of ZnO nanostructure was not affected by UV irradiation to the suspension as shown in XRD patterns of Figure S2. However, the amount of metals deposited on the ZnO nanostructures would be increased with increasing the UV intensity and deposition time, and then saturated. Since the gas sensing response of metal/semiconductor sensor would be closely related to the amount of deposited metal, the effect of UV intensity and deposition time (irradiation time) upon the amount of metal deposited on ZnO nanostructures was checked using 0.1 mM of metal precursors to optimize the deposition amount of each metal. The deposition amount of metals increased with increasing the UV intensity up to 450 W for 30 min in deposition process. The content of deposited metal was observed to be saturated at condition over 30 min. On the basis of this result, we set the experimental condition for the deposition of metal as UV intensity of 450 W for 30 min in order to obtain the best gas response throughout the optimal loading content of metal nanoparticles deposited on ZnO nanostructures.Therefore, after ~
Figure S2. XRD patterns of the samples
Comment 2. The paper uses the XPS method. The Au and Cu spectra are discussed and the conclusion about the electronic state of these elements is made. Obviously, the authors should have similar data for oxygen (O1s). It is necessary to add these results to the article and discuss how the amount of surface oxygen changes depending on the microstructure of zinc oxide particles and the presence of deposited copper and gold particles.
Response 2.
We agree with the reviewer’s opinion that the change of surface oxygen components in the samples is one of important factors to evaluate the gas sensitivity of the sensor. Therefore, we analyzed the XPS O1s data of the samples. According to the XPS O1s data of the samples, It was confirmed that the amount of surface oxygen changed depending on the morphology of ZnO and the kinds of the deposited metals on the microstructures of ZnO. There were three kinds of surface oxygen in our samples, lattice oxygen, oxygen vacancy, and chemisorbed oxygen which were assigned at 530.3, 531.2, and 532.6 eV, respectively in XPS O1s spectra. As shown in Table S1, it was observed that oxygen vacancies were more created in ZnO nanorod than in ZnO nanoplate. In addition, it was observed that the deposition of the metal increases the creation of the oxygen vacancies in the samples. Among the samples, Au/ZnO nanorod exhibited the largest portion of oxygen vacancies, which was presumed to be helpful in the response to the CO gas.
Related discussions are added in the Results and discussion as follows.
Page 9, Line 291
~ exposure of CO gas. In general, it was well known that the surface oxygen of the sample also influences the gas response. Therefore, we analyzed the amount of surface oxygen of the sample by XPS analysis. Figure 8 shows the XPS O1s spectra of the ZnO nanostructures and metal-deposited ZnO structures. It was found that the samples were composed of three kinds of surface oxygen with different fractions. The peaks in XPS O1s spectra assigned at 530.3, 531.2, and 532.6 eV were lattice oxygen, oxygen vacancy, and chemisorbed oxygen, respectively. As shown in Table 4, it was observed that oxygen vacancies were more created in ZnO nanorod than in ZnO nanoplate. In addition, it was observed that the deposition of the metal increases the creation of the oxygen vacancies in the samples. Therefore, it was confirmed that the morphology of ZnO nanostructures and the deposition of metal on the ZnO nanostructures affect the amount of surface oxygen of the samples. Among the samples, Au/ZnO nanorod exhibited the largest portion of oxygen vacancies, which was also believed to have contributed to the response to the CO gas. The response time ~
Figure 8. XPS O1s spectra of the ZnO nanostructures and metal-deposited ZnO nanostrucrtures (a) ZnO nanorod (b) ZnO nanoplate (c) Au/ZnO nanorod (d) Au/ZnO nanoplate (e) Cu/ZnO nanorod (f) Cu/ZnO nanoplate
Table 4. Surface oxygen components in ZnO nanostructures and metal-deposited ZnO nanostructures.
|
Sample |
OL1(%) |
Ov2(%) |
Oc3(%) |
|
ZnO nanorod |
39.63 |
46.39 |
13.98 |
|
Au/ZnO nanorod |
43.49 |
51.66 |
4.85 |
|
Cu/ZnO nanorod |
52.78 |
43.91 |
3.31 |
|
ZnO nanoplate |
62.59 |
31.47 |
5.94 |
|
Au/ZnO nanoplate |
50.00 |
48.31 |
1.69 |
|
Cu/ZnO nanoplate |
54.20 |
45.60 |
0.20 |
1 lattice oxygen; 2 oxygen vacancy; 3 chemisorbed oxygen.
Comment 3. In addition, it would be useful to make a blank experiment in which the conditions for the deposition of metal particles would be reproduced, without introducing metal precursors. The ZnO particles treated in this way should also be examined using the XPS method. In this way, it is possible to evaluate the effect of UV radiation directly on the characteristics of zinc oxide particles.
Response 3.
We think that reviewer’s comment about blank experiment is a very helpful suggestion, although, regretfully,we didn’t carry out the blank experiment. In this point of view, it was very sorry that direct evaluation throughout blank experiment could not be suggested. Instead, it might be indirectly inferred that the crystal phase and morphological characteristics of ZnO nanostructures would be rarely affected by UV radiation from comparison of XRD (Figure S2) and SAED pattern (onset of Figure 2) between as-prepared ZnO nanostructures and metal-deposited ones which were irradiated by UV during process of metal deposition, since the as-prepared ZnO nanostructures and metal-deposited ones showed almost the same crystal pattern of ZnO. From the result that there is no changes in crystal phase and morphology of ZnO after photodeposition of the metals, it might be suggested that UV irradiation wouldn’t affect ZnO nanostructures severely.
Comment 4. The authors presented very interesting results on the dynamic characteristics of sensors - a significant reduction in response time and recovery time in the case of modified samples compared to pure ZnO. This fact requires a deeper discussion.
Response 4.
Firstly, we are very appreciated to this reviewer’s comment about response time and recovery time because it was very interesting result that was unintentionally obtained. In this study, in order to investigate the effect of morphology and electrical property of sensing material upon gas response, we have prepared ZnO nanostructures with two different morphology of nanorod and plate by controlling synthetic condition and deposited metal nanoparticles on ZnO nanostructures through photodepositon. This effect of morphology and metal deposition upon gas response, working temperature and so on was confirmed and discussed in this work, but, unfortunately, the origin of the shorter response time and recovery time couldn’t be still explained fully. Although it was guessed that the improved electrical conductivity by metal deposition might have a little influence on such shorter response and recovery time, it was concluded the result could not be understood due to such factor alone. As reviewer pointed out, we also thought that a significant reduction in response time and recovery time in the case of modified samples compared to pure ZnO would be a very interesting and important subject and need to be studied throughout the further experiment and research.
Comment 5. The title of the article is "Effect of the morphology...", but in fact this effect is only postulated on the basis of experimental data, but there are no theoretical or model explanations for this effect in the work. This is the weak point of the reviewed article.
Response 5.
We agreed with reviewer’s comment saying that the theoretical or model calculation and explanations about the effect of morphology and electrical property of sensing materials on gas sensing properties were insufficient, and are sorry for that. As stated in response 4, we have studied for the purpose of experimentally elucidating the effects of some factors such as morphology, electrical property and so on upon gas sensing property. Instead, according to reviewer’s kind comment, brief explanation about the improved gas response in Au/ZnO and Cu/ZnO relative to pure ZnO was suggested based on the band alignment between the metal and semiconductor used herein as shown in Figure 6.
Related discussion and Figure was added in Result and Discussion part of revised manuscript as follows
Figure 6. (a) Energy levels of ZnO and Au; (b) formation of heterojunction barrier. (c) Energy levels of ZnO and Cu; (d) formation of heterojunction barrier.
Page 8, Line 273
According to the gas-sensing principle, we can predict that a better response can be achieved by making a larger change of work function or electrical conductivity of the sensor under the exposure of air and CO. In this point of view, the metal-deposited ZnO nanostructures, due to the larger electrical conductivity and higher work function of deposited-metal relative to ZnO, could bring about a larger difference in the height of the Schottky barrier, leading to show the improved gas response [29,30]. It was found in Table 1 that the electrical conductivity of the samples became higher in the order of Au-deposited ZnO nanostructures, Cu-deposited ones, and pristine ones due to the actual fraction of metal-deposited on ZnO nanostructures. The work function of the deposited-metal became also higher in the order of Au(5.10 ~ 5.47 eV) and Cu(4.53 ~ 5.10 eV), which was reported elsewhere. As shown in Figure 6, due to the higher work function of Au relative to ZnO, electrons migrate from the conduction band of ZnO to Au in order to equalize the Fermi levels and form Schottky heterojunctions at the interface of Au and ZnO nanostructures, which leading to the reduction of conduction volume. Such reduction of conduction volume will contribute to the enhanced sensor response in Au/ZnO [29]. In analogous to the case of Au/ZnO, such reduction in Cu/ZnO will also occur but be smaller than that in Au/ZnO due to the lower work function of Cu relative to Au. Therefore, we can expect that the Au-deposited ZnO nanorods, which have the highest electrical conductivity and work function of deposited-metal, would elicit the best response to the exposure of CO gas.~

Reviewer 2 Report
Please refer to the attached file.

Author Response
Dear Reviewers
First of all, I’d like to express my sincere and thanks to the Reviewers who read my manuscript with deep concerns and for valuable comments.
According to the reviewer’s comments, the manuscript was revised and checked to avoid some typical mistakes and expression or notation to cause misunderstanding.
We appreciate the kind comments and suggestions of the reviewer again. Our responses to these comments are listed below, and the manuscript has been accordingly revised. All the changes were represented in blue color in the revised manuscript.
Thank you for your kindness and help
Sincerely yours
Response to Reviewer 2 comments
The submitted work reports preparation of metal-deposited ZnO nanorod and nanoplate via photodeposition. The CO gas sensing performances of the Au/ZnO and Cu/ZnO are compared in this study. The TEM works show the local microstructures of the prepared metal/ZnO composites. The XPS results demonstrate the copper and gold binding states. Finally, the authors concluded that the CO gas sensitivity of the prepared samples is influenced by the combined effect such as the work function of deposited-metal and electrical conductivity, the amounts and type of the deposited-metal nanoparticles, and the morphology of the ZnO nanostructures. In fact, many works on gas sensors made from metal/semiconductor composites have been reported before. The authors need to highlight the novelty and importance of the submitted work. Moreover, there are several drawbacks need to be further improved before this manuscript can be accepted for publication.
Comment 1. First, the authors need to compare literatures with their work and explain why the metal particle decoration process used herein is more advantageous than the reported works using other synthesis routes.
Response 1.
First, thank you for kind comment. According to the reviewer’s comment, we added some literatures about the previously reported metal/semiconductor gas sensors and some advantages of photo deposition method used in this work in introduction section of revised manuscript as follows
Related discussion is added in the introduction part as follows.
Page 2, Line 47
~ compared. Although there are several works reported about synthetic routes to prepare the metal/semiconductor gas sensors via chemical doping [10] or chemical reduction of metal ion [11-13] as listed in Table 1, to our knowledge, there are barely reported about metal/semiconductor gas sensor prepared via photodeposition. The photodeposition method has some more advantages than other methods, since metal nanoparticles could be not only deposited on the substrate in uniform size and distribution by photodeposition, but also the fabrication process would be easier and more simple than other synthetic routes. ~
Table 1. Comparison of the CO sensitivity of metal/semiconductor gas sensors with different synthetic routes.
|
Nano particles |
Morphology (ZnO) |
Synthesis method |
Operating temperature (°C) |
[CO] (ppm) |
Sensitivity (Ra/Rg) |
Ref. |
|
|
In |
nanoparticle |
Sol-gel |
300 |
100 |
5 |
[10] |
|
|
Au |
Nanostar |
Hydrothermal |
RT |
100 |
15 |
[12] |
|
|
Pt |
nanosheet |
Hydrothermal/ calcination |
180 |
100 |
3.5 |
[11] |
|
|
Au |
nanorods |
Hydrothermal |
150 |
1000 |
12 |
[13] |
|
|
Pd |
nanowires |
VLS growth |
RT |
0.1 |
1.02 |
[13] |
|
Comment 2. Why the authors choose Cu particle as a control sample? The works on gas-sensing performance of Cu-decorated ZnO (or other semiconductors) should be reviewed and commented in the introduction section.
Response 2.
In our experiment, the electrical property of the metal nanoparticles was one of important variables to affect the CO gas response. In order to investigate such effect, generally noble metal would be considered as the best candidates due to their high electrical conductivity and stability. In this work, considering practical application, we wanted to find other metal of high electrical conductivity and stability with the price competitiveness to alternate noble metals. In this aspect, we chose Cu nanoparticles as one of the candidate materials to alternate noble metals. It was reported that the Cu particle plays a very important role in the CO sensing saying that the coexistence of ZnO and Cu metal could enhance the capability of sensor material to adsorb CO molecules since the copper site acts as active one to adsorb CO molecules at both low and high temperature.
Related discussion is added in the introduction part as follows.
Page 2, Line 53
The electrical property of the metal nanoparticles might be one of important variables to affect the CO gas response. In order to investigate such effect, generally noble metal would be considered as the best candidates due to their high electrical conductivity and stability. Furthermore, in this work, considering practical application, we wanted to find other metal of high electrical conductivity and stability with the price competitiveness to alternate noble metals. In this aspect, altogether with Au nanoparticle, Cu one was chosen as one of the candidate materials. Interestingly, it was reported that both Au and Cu nanoparticles have important role in CO sensing [14,15]. The presence of Au nanoparticles on the surface of In2O3 nanowire serves to enhance the CO oxidation due to a higher oxygen ion-chemisorption on the conductive Au nanoparticles surfaces. The coexistence of ZnO and Cu metal could enhance the capability of sensor material to adsorb CO molecules since the copper site acts as active one to adsorb CO molecules at both low and high temperature [15]. In this study, ~
Comment 3. What is the reason for using ZnOs in the forms of rod and plate to act as templates for sensing materials? ZnO with these two morphologies have different specific surface areas. Therefore, different gas-sensing performances of these two types of ZnO templates are predictable.
Response 3.
As reviewer said, we used ZnO nanostructures in two different forms with expecting the different specific surface area and dimensional characteristics. That is, one of the objectives of this study was to investigate how the CO gas sensitivity is affected by the surface properties of ZnO nanostructures. To clarify the difference in the surface properties of ZnO, we selected two nanostructures with different morphology (rod and plate). The results showed that the nanorods showed better CO gas sensitivity than the nanoplates due to the higher specific surface area of nanorods as expected from difference in morphology such as size and shape of ZnO particles.
Comment 4. The authors only demonstrated TEM results to show microstructures of the metal/ZnO samples. The overall microstructures of the prepared samples characterized by XRD and SEM are necessary to understand the uniformity of the metal particle–decorated nanostructured samples.
Response 4.
As the reviewer pointed out, we had presented only TEM images to show the microstructures of metal/ZnO nanostructures. And so, now, according to reviewer’s comment, the related discussion and SEM and XRD data were added in revised manuscript. In revised manuscript, both ZnO nanorod and nanoplate are found to exhibit the uniform overall morphology and distribution from SEM images of Figure S1. On the other hand, to investigate the phase of metal-deposited ZnO nanostructures, we carried out XRD and SAED patterns analysis. XRD in Figure S2 showed that only the crystal phase of ZnO alone was observed in all the samples due to the low crystallinity and the deposition amount of small metal nanoparticle (about under 10nm). TEM and SAED patterns in Figure 2 showed the local microstructure and crystal phase of metal and ZnO nanoparticle in all the metal-deposited ZnO nanostructures.
Related discussions are in the result and discussion section and some figures in supplementary information as follows.
Page 5, Line 146
The local microstructure of all the samples were showed in Figure 2, in which nanorods were much smaller than nanoplates and very small metal nanoparticles such as copper and gold nanoparticles were observed to be deposited on ZnO nanorods and nanoplates. In the case of overall morphology, both ZnO nanorod and nanoplate were observed to exhibit the uniform overall morphology and distribution (SEM images of Figure S1). On the other hand, to investigate the phase of metal-deposited ZnO nanostructures, we carried out XRD and SAED patterns analysis. While XRD in Figure S2 showed that only the crystal phase of ZnO alone was observed in all the samples due to the low crystallinity of small metal nanoparticles (~ 10nm) and smaller content of deposited-metal for that of ZnO, it could be confirmed that, from TEM images and onset SAED patterns of Figure 2, the deposited-metal species were stabilized on ZnO nanostructures as metal phase. On the other hand, it seemed in Figure 2 that a larger amount ~
Figure S1. SEM image of ZnO nanostructures (a) ZnO nanorod (b) ZnO nanoplate
Figure S2. XRD patterns of the samples
Comment 5. The loading content of the metal particles is an important factor that dominates the gas-sensing performance. What is the optimal loading content of Au and Cu particles on the ZnO? How the CO gas-sensing response varies with the loading content of the metal particles herein?
Response 5.
As reviewer pointed out, the loading content of the metal particles is a very important factor that dominates the gas sensing performance since the gas sensing response of metal/semiconductor sensor was known to be improved in proportional with the loading content. In this work, firstly, we examined the effect of UV intensity and deposition time (irradiation time) upon the content of metal deposited on ZnO nanostructures using 0.1 mM of metal precursors to optimize the deposition amount of each metal. The deposition amount of metals increased with increasing the UV intensity up to 450 W for 30 min in deposition process. The content of deposited metal was observed to be saturated at condition over 30 min. Therefore, we set the experimental condition for the deposition of metal as UV intensity of 450 W for 30 min in order to obtain the best gas response throughout the optimal loading content of metal nanoparticles deposited on ZnO nanostructures, and then characterized samples prepared under such photodeposition condition.
Related discussion was added in experimental section (2. Materials and Methods) of revised manuscript
Page 3, Line 90
~ respectively. It was found that the crystal phase of ZnO nanostructure was not affected by UV irradiation to the suspension as shown in XRD patterns of Figure S2. However, the amount of metals deposited on the ZnO nanostructures would be increased with increasing the UV intensity and deposition time, and then saturated. Since the gas sensing response of metal/semiconductor sensor would be closely related to the amount of deposited metal, the effect of UV intensity and deposition time (irradiation time) upon the amount of metal deposited on ZnO nanostructures was checked using 0.1 mM of metal precursors to optimize the deposition amount of each metal. The deposition amount of metals increased with increasing the UV intensity up to 450 W for 30 min in deposition process. The content of deposited metal was observed to be saturated at condition over 30 min. On the basis of this result, we set the experimental condition for the deposition of metal as UV intensity of 450 W for 30 min in order to obtain the best gas response throughout the optimal loading content of metal nanoparticles deposited on ZnO nanostructures.Therefore, after ~
Comment 6. No difference in optimal working temperature for the Au/ZnO and Cu/ZnO, and also for the ZnO rod and ZnO plate. Why?
Response 6.
As reviwer said, it was observed that optimal working temperature was affected mainly by metal deposition, while rarely by morphology or deposited-metal species. As already stated in manuscript, since the optimal working temperature was known to be closely related with adsorption-desorption of oxygen species such as monovalent or divalent oxygen, surface electrostatic state of particles might became more important factor. And so it can be expected that the decoration of semiconductor with metal which could lead to the change of surface electrical characteristics and the lowering of the activation energy for chemisorption of CO gas, might affect the optimal working temperature much more largely rather than the change in morphology such as shape or size of particles which was difficult to induce a distinguishable difference in surface electric state.
Comment 7. The gas-sensing selectivity of the metal-particle decorated oxides has been reported. However, the authors did not show gas-sensing selectivity of the prepared samples, weakening the potential sensor device applications. Please provide cross-sensitivity of their samples over several reducing and oxidizing target gases.
Comment 8. The humidity is also an important factor to influence the CO gas-sensing performance of metal/oxide composites. The authors should address this issue in their work to improve the scientific quality of the manuscript.
Response 7 and 8.
As reviewer pointed out, the gas-sensing selectivity and the effect of humidity are also very important subject, especially for industrial application. However, unfortunately, such research and experiment about factors such as the selectivity or the effect of humidity were not carried out in this study, because the main objective in this our work was to investigate the correlation between the CO gas sensitivity and the morphological and electrical properties of the ZnO and so we focused on investigating the factors for lowering the operating temperature of the CO gas sensor and improving the sensitivity by changing the morphology of ZnO and kinds of deposited metals. ZnO gas sensor is reported to be used as gas sensors for various target gases such as CO, H2S, Ethanol, NH3, HCHO, etc as reviewer said. Among them, we selected CO gas known as a representative target gas of ZnO sensor to achieve our objective of the research. In this work, the gas sensing test was performed under vacuum chamber, in which the relative humidity can be considered to be almost zero. However, as stated above, since factors such as the gas-sensing selectivity and humidity are very important for evaluating the gas sensing performance and practical application of the sensor, the further research about such subject is thought to be required for wide application.
Comment 9. Is that possible for the Cu particles oxidized during the gas-sensing tests? Please provide the proof of the Cu binding energy after cyclic sensing measurements.
Response 9. From the result of cyclic sensing measurement, gas response was confirmed to be stable even after 10 times successive cyclic test, which could lead to the inference that Cu nanoparticles would remain stably in metal phase, with not oxidized during the gas-sensing test. According to reviewer’s comment, Cu and Au XPS spectra for Cu/ZnO nanorod and nanoplate and Au/ZnO ones before and after gas-sensing tests were presented in revised manuscript. As expected from cyclic measurement, Cu 2p spectra show no much difference in the chemical state of Cu nanoparticles on Cu/ZnO nanorod and Cu/ZnO nanoplate before and after gas sensing test.
Related discussion and Figure were added in the results and discussion part of revised manuscript.
Page 11, Line 325
~ reversible cycles. As shown in Figure 10, we also investigated the change of chemical state of metal in the metal-deposited ZnO nanostructures after 10 successive cycles of exposure to 100 ppm CO gas. Au 4f and Cu 2p spectra of samples before and after gas-sensing test didn’t show any prominent difference, reflecting that there was no change in the chemical state of Au and Cu nanoparticles after gas-sensing test. Those results would also indicate that the metal-deposited ZnO nanostructures have good stability.
Figure 10. The chemical states of metal deposited ZnO structures before and after the stability test. (a) Au 4f spectra of Au/ZnO nanorod (b) Au 4f spectra of Au/ZnO nanoplate (c) Cu 2p spectra of Cu/ZnO nanorod (d) Cu 2p spectra of Cu/ZnO nanoplate.
Comment 10. It would be better to show the schematic of the sensor device used herein for clarity.
Response 10.
The schematic of the sensor device used herein is added in the experimental section.
Figure 1. The schematic illustration of the sensor device.
Comment 11. The CO gas sensing mechanisms of Au/ZnO and Cu/ZnO should be provided and discussed based on the band alignment between the metal and semiconductor used herein.
Response 11.
According to reviewer’s comment, the improved CO gas-sensing mechanism of Au/ZnO and Cu/ZnO were suggested and discussed in revised manuscript as follows.
Related discussion and Figure were added in the results and discussion of revised manuscript as follows.
Figure 6. (a) Energy levels of ZnO and Au; (b) formation of heterojunction barrier. (c) Energy levels of ZnO and Cu; (d) formation of heterojunction barrier.
Page 8, Line 273
According to the gas-sensing principle, we can predict that a better response can be achieved by making a larger change of work function or electrical conductivity of the sensor under the exposure of air and CO. In this point of view, the metal-deposited ZnO nanostructures, due to the larger electrical conductivity and higher work function of deposited-metal relative to ZnO, could bring about a larger difference in the height of the Schottky barrier, leading to show the improved gas response [29,30]. It was found in Table 1 that the electrical conductivity of the samples became higher in the order of Au-deposited ZnO nanostructures, Cu-deposited ones, and pristine ones due to the actual fraction of metal-deposited on ZnO nanostructures. The work function of the deposited-metal became also higher in the order of Au(5.10 ~ 5.47 eV) and Cu(4.53 ~ 5.10 eV), which was reported elsewhere. As shown in Figure 6, due to the higher work function of Au relative to ZnO, electrons migrate from the conduction band of ZnO to Au in order to equalize the Fermi levels and form Schottky heterojunctions at the interface of Au and ZnO nanostructures, which leading to the reduction of conduction volume. Such reduction of conduction volume will contribute to the enhanced sensor response in Au/ZnO [29]. In analogous to the case of Au/ZnO, such reduction in Cu/ZnO will also occur but be smaller than that in Au/ZnO due to the lower work function of Cu relative to Au. Therefore, we can expect that the Au-deposited ZnO nanorods, which have the highest electrical conductivity and work function of deposited-metal, would elicit the best response to the exposure of CO gas.~

Reviewer 3 Report
- Data provided by the authors is good but for a sensor to be unique, SELECTIVITY of the sensor should be tested with different gases. If possible, addition of selectivity of the gas sensor can be added to the present work.
- Does your sensor have a saturation limit after 100ppm?
- Since your work contains 2 different metals deposited on ZnO nanostructure, gas sensing mechanism could have explained with detailed reactions for more detailed understanding of the readers. I suggest you to cite the reference of “Au doping ZnO nanosheets sensing properties of Ethanol gas prepared on MEMS device, Coatings 10 (2020) 945.”
Author Response
Dear Reviewers
First of all, I’d like to express my sincere and thanks to Reviewers who read my manuscript with deep concerns and to valuable comments.
According to reviewer’s comments, manuscript was revised and checked to avoid some typical mistakes and expression or notation to cause misunderstanding.
We appreciate kind comments and suggestion of reviewer again. Our responses to these comments are listed below, and the manuscript have been accordingly revised. All the changes were represented in blue color in revised manuscript.
Thank you for your kindness and help
Sincerely yours
Response to Reviewer 3 comments
Comment 1. Data provided by the authors is good but for a sensor to be unique, SELECTIVITY of the sensor should be tested with different gases. If possible, addition of selectivity of the gas sensor can be added to the present work.
Response 1.
As reviewer pointed out, the selectivity of sensor is very important subject, especially for industrial application. However, unfortunately, research and experiment about the selectivity were not carried out in this study, because the main objective in this our work was to investigate the correlation between the CO gas sensitivity and the morphological and electrical properties of the ZnO and so we focused on investigating the factors for lowering the operating temperature of the CO gas sensor and improving the sensitivity by changing the morphology of ZnO and kinds of deposited metals. ZnO gas sensor is reported to be used as gas sensors for various target gases such as CO, H2S, Ethanol, NH3, HCHO, etc as reviewer said. Among them, we selected CO gas known as a representative target gas of ZnO sensor to achieve our objective of the research. However, as stated above, since the gas-sensing selectivity is very important for evaluating the gas sensing performance and practical application of the sensor, the further research about such subject is thought to be required for wide application.
Comment 2. Does your sensor have a saturation limit after 100 ppm?
Response 2.
Unfortunately, in this work, we didn’t carry out the experiment about saturation limit. However, according to the reported literature, a saturation limit of CO gas is found to be around 2000 ppm [1].
- Neri, G.; Bonavita, A.; Micali, G.; Rizzo, G.; Callone, E.; Carturan, G. Resistive CO gas sensors based on In2O3 and InSnOx nanopowders synthesized via starch-aided sol–gel process for automotive applications. Sens. Actuators, B 2008, 132, 224-233, doi:https://doi.org/10.1016/j.snb.2008.01.030.
Comment 3. Since your work contains 2 different metals deposited on ZnO nanostructure, gas sensing mechanism could have explained with detailed reactions for more detailed understanding of the readers. I suggest you to cite the reference of “Au doping ZnO nanosheets sensing properties of Ethanol gas prepared on MEMS device, Coatings 10 (2020) 945.”
Response 3. Thank you for your suggestion. The reference was cited in revised manuscript to explain the CO gas sensing mechanism of metal deposited ZnO nanostructures. It would be very helpful to understand the gas sensing mechanism.
Related discussions are in the results and discussion as follow.
Page 8, Line 250
~ ZnO is an n-type semiconductor with contains free electrons, after photodeposition of metal on the ZnO nanostructures, free electrons from metal are released into the ZnO structure and the number of free electrons will be more [28]. According to the principle of CO gas sensing, the oxygen species are adsorbed on the surface of the n-type semiconductor gas sensor, which results in a high resistance in the atmosphere due to the formation of a depletion layer. After exposure to CO gas, the adsorbed oxygen species reacted with CO molecules, which releases the free electrons back to the ZnO structure, and ~

Round 2
Reviewer 1 Report
In the current version of the article, the interpretation of XPS oxygen spectra is not entirely correct. The authors attribute a peak in the O1s spectrum with an energy of 531.2 eV to oxygen vacancies. Taking into account the physical basis of the XPS method, this interpretation is erroneous, since the vacancy corresponds to the absence of an oxygen atom in its position of the crystal structure. Therefore, it is impossible to get a signal from the photoelectron of an oxygen atom if this atom is absent.
Moreover, I didn't notice that the authors used the Shirley method for the background. This may be the reason for the deconvolution of the spectrum into three components. In all the spectra, except for the one shown in Fig. 8A, the peak with an energy of 532.6 eV is very weak. Since this peak has an exceptionally low intensity, I propose to describe the O1s spectra with only two components corresponding to lattice and adsorbed oxygen.
Author Response
Responses to the Reviewers
Manuscript ID: Nanomaterials-966283
Title: Effect of the morphology and electrical property of metal-deposited ZnO nanostructures on CO gas sensitivity
Dear Reviewers
First of all, I’d like to express my sincere and thanks to the reviewers who read my manuscript with deep concerns and for valuable comments.
According to the reviewer’s comments, the manuscript was revised and checked to avoid some typical mistakes and expression or notation to cause misunderstanding.
We appreciate the kind comments and suggestions of the reviewer again. Our responses to these comments are listed below, and the manuscript has been accordingly revised. All the changes were represented in blue and red colors in the revised manuscript.
Thank you for your kindness and help
Sincerely yours
Response to Reviewer 1 comments
Comment 1.
In the current version of the article, the interpretation of XPS oxygen spectra is not entirely correct. The authors attribute a peak in the O1s spectrum with an energy of 531.2 eV to oxygen vacancies. Taking into account the physical basis of the XPS method, this interpretation is erroneous, since the vacancy corresponds to the absence of an oxygen atom in its position of the crystal structure. Therefore, it is impossible to get a signal from the photoelectron of an oxygen atom if this atom is absent.
Moreover, I didn't notice that the authors used the Shirley method for the background. This may be the reason for the deconvolution of the spectrum into three components. In all the spectra, except for the one shown in Fig. 8A, the peak with an energy of 532.6 eV is very weak. Since this peak has an exceptionally low intensity, I propose to describe the O1s spectra with only two components corresponding to lattice and adsorbed oxygen.
Response 1.
Thanks very much for the kind and helpful review and sorry to confuse you. If you hadn't pointed out, our mistakes would have made readers badly misunderstood. As you pointed out, the vacancy, which corresponds to the absence of an oxygen atom in its position of the crystal structure, is impossible to get a signal since it means that there is no atom. The expression ‘vacancy oxygen’ in the discussion about XPS of the manuscript was our mistake. The correct expression is ‘oxygen ions in the oxygen-deficient region caused by oxygen vacancy’. According to literature (M. Chen et al, Applied Surface Science, and so on), the peak of surface O1s XPS spectra was reported to be separately assigned to three-component. The reports said that the high binding energy component located at ca. 532.4 eV is usually attributed to the presence of loosely bound oxygen (chemisorbed oxygen) belonging to a specific species, e.g., adsorbed CO2, H2O, or O2, the one on the low binding energy side of the O1s spectrum at ca. 530.15 eV is to O2- ions (lattice oxygen) on wurtzite structure of hexagonal Zn2+ ion array, surrounded by Zn atoms with their full complement of nearest-neighbor O2- ions, and finally, the medium binding energy one, centered at ca. 531.25 eV, is associated with O2- ions in the oxygen-deficient regions within the matrix of ZnO. And so, on the basis of these reports about O1s XPS of ZnO, the peak of O1s XPS spectra in our study was fitted by three nearly Gaussian functions, centered at 530.15, 531.25, and 532.40 eV, respectively as the initial value for fitting.
- cf) The literature referenced to assign O1s XPS peaks
- Chen, X. Wang, Y.H. Yu, Z.L. Pei, X.D. Bai, C. Sun, R.F. Huang, L.S. Wen, Applied Surface Science 158 _2000. 134
- Major, S. Kumar, M. Bhatnagar, K.L. Chopra, Appl. Phys.Lett. 40 _1986. 394.
M.N. Islam, T.B. Ghosh, K.L. Chopra, H.N. Acharya, Thin Solid Films 280 _1996. 20.
- Cebulla, R. Weridt, K. Ellmer, J. Appl. Phys. 83 _1998. 1087.
- Szore´nyi, L.D. Laude, I. Bertoti, Z. Ka´ntor, Z. Ge´re-tovszky, J. Appl. Phys. 78 _1995. 6211.
L.K. Rao, V. Vinni, Appl. Phys. Lett. 63 _1993. 608.
J.C.C. Fan, J.B. Goodenough, J. Appl. Phys. 48 _1977. 3524.
According to the reviewer’s comment, we revised the incorrect assignment in O1s XPS spectra to the correct one.
We are sorry again for troubling you.
Related corrections are in the results and discussion as follows.
Page 9, Line 288
~To estimate the fraction of each component of surface oxygen, the XPS O1s spectra were deconvoluted using the Shirley method. Figure 8 shows the XPS O1s spectra of the ZnO nanostructures and metal-deposited ZnO structures. It was found that the samples were composed of three kinds of surface oxygen with different fractions. The peaks in XPS O1s spectra assigned at 530.3, 531.2, and 532.6 eV were lattice oxygen, oxygen ions in the oxygen-deficient regions caused by oxygen vacancy, and chemisorbed oxygen, respectively [31,32]. As shown in Table 5, it was found that the peak at ca. 531.2 eV attributed to oxygens in oxygen-deficient regions oxygen vacancies were larger in ZnO nanorod than in ZnO nanoplate, reflecting more oxygen vacancies in ZnO nanorod than in ZnO nanoplate. In addition, except for Cu/ZnO nanostructure, it was expected from the larger peak area observed in metal-deposited ZnO nanostructures than pure ones that the deposition of metal nanoparticles, especially Au nanoparticles, also increases the creation of the oxygen vacancies in the samples. Therefore~
Page 10, Line 310 footnote at Table 5
1 lattice oxygen; 2 oxygen around vacancy; 3 chemisorbed oxygen.

Reviewer 2 Report
I am satisfied with the quality of the revised version. The revised version can be accepted for publication in its current form.
Author Response
Responses to the Reviewers
Manuscript ID: Nanomaterials-966283
Title: Effect of the morphology and electrical property of metal-deposited ZnO nanostructures on CO gas sensitivity
Dear Reviewers
I’d like to express my sincere and thanks to the reviewers who read my manuscript with deep concerns and for valuable comments.
Thank you for your kindness and help.
Sincerely yours,
Reviewer 3 Report
The manuscript is suitable to publish at this Journal.
Author Response

(The authors gave the same response as above.)
